# Antimicrobial Activity and Cytotoxicity of Prepolymer Allyl 2-cyanoacrylate and 2-Octyl Cyanoacrylate Mixture Adhesives for Topical Wound Closure

**DOI:** 10.3390/ma16093427

**Published:** 2023-04-27

**Authors:** Soyeon Oh, Dae-Hyun Hahm, Yong-Bok Choi

**Affiliations:** 1Theramx Inc., Starwood, Jungwon-gu, Seongnam-si 13229, Republic of Korea; 2Department of Physiology, College of Medicine, Kyung Hee University, Seoul 02447, Republic of Korea

**Keywords:** antimicrobial, prepolymer allyl 2-cyanoacrylate, 2-octyl cyanoacrylate, skin adhesive, wound closure, biocompatibility

## Abstract

The development of a new skin adhesive that can be used inside and outside the body, which prevents infection and has fewer scars and less side effects, is currently attracting attention from the scientific community. To improve biocompatibility, prepolymer allyl 2-cyanoacrylate (PAC) and 2-octyl cyanoacrylate (OC) were mixed in various proportions and tested for their therapeutic potential as skin adhesives. A series of skin adhesive samples prepared by mixing PAC, OC, and additives with % (*w*/*w*) ratios of 100:0:0, 0:100:0, 70:0:30, 40:30:30, and 30:40:30 were tested to determine their antimicrobial activity, cell cytotoxicity, and formaldehyde release. The additives include myristic acid and dibutyl sebacate as plasticizers and butylated hydroxyanisole as an antioxidant. It was observed that the samples containing 70% PAC (PAC7) or 40% PAC (PAC4) with 30% additives had the highest antimicrobial activities against various microbial cells and no cytotoxicity regarding in vitro fibroblast cell growth. In addition, these formulations of adhesive samples released formaldehyde within the levels permitted for medical devices. Taken together, the mixture of PAC and OC as a topical skin adhesive for wound closure was found to be biocompatible, mechanically stable and safe, as well as effective for wound healing.

## 1. Introduction

A skin wound provides a growth surface and an ample supply of nutrients for bacterial infection, colonization, and biofilm formation, which can lead to sepsis [1]. Therefore, protecting the wound from bacterial infection and absorbing excess exudate in the wounded area should be achieved quickly. Choosing the correct dressing is an important factor that reduces the healing time, provides cost-effective care, and improves the patient’s quality of life, as well as preventing infection [2]. Thus, researchers are constantly exploring new materials to overcome the significant limitations of wound dressings, such as the risk of immune rejection and toxicity, loss of integrity of the dressing, and excessive swelling due to the accumulation of sub-membrane fluid. Various types of tissue adhesives have been developed, and new materials used to promote wound healing are gaining increased popularity in diverse clinical applications [2,3].

In recent years, one of the most significant discoveries in wound closure materials has been cyanoacrylate polymer adhesives [4], which have been widely used as biological adhesives that maintain the healing environment while protecting the wound from external pathogens, preventing bacterial infection in the wounded area that may result from exposure to the external environment [5,6]. Among the various types of cyanoacrylate polymer adhesives, alkyl cyanoacrylate adhesives and their formulated compounds achieve the fastest polymerization via implementation throughout any substratum composed of moisture, even under ambient circumstances, without the need for any internal source to be healed [5,7]. Alkyl cyanoacrylate adhesives can cure the wound in ambient conditions without the aid of an external energy source, and they are also economical to use due to the small amount required for a bond [8]. Several types of alkyl cyanoacrylate adhesives are now being sold as retail and industrial adhesives globally [9]. However, numerous studies have shown that traditional cyanoacrylate is used only as a temporary adhesive due to its poor mechanical properties and local tissue toxicity [10,11]. Alkyl octyl cyanoacrylate adhesives with a side chain of eight carbon atoms have exhibited themselves as significant advances in terms of mechanical properties in the field of wound closure adhesive materials [12]. A longer side chain of octyl cyanoacrylate provides various potential benefits, such as a greater strength and flexibility to cyanoacrylate adhesives than those of medium- or short-chained cyanoacrylate adhesives, including butyl cyanoacrylate adhesives. It also exhibited four times the three-dimensional breaking strength. However, even though OC adhesives are commonly used as one of the largest bandage products ranked by dollar revenues in the US, their use has been restricted because of its low peel and impact strength [13,14].

Recently, an advanced cyanoacrylate, pre-polymerized allyl 2-cyanoacrylate, has been manufactured for applications as bio-glue through the partial polymerization of the intramolecular double bonds, whereby the partial polymerization of the intramolecular double bonds of PAC helps to increase the stability and biocompatibility. PAC is produced as a prepolymer with an excellent binding force and biocompatibility [10,11,15]. Despite its ability to enhance the mechanical properties of the adhesive, such as its bonding strength, its usage in medicine is still limited due to a lack of awareness in the industry and its high cost [16,17,18].

In order to improve the mechanical properties, biocompatibility, and production costs simultaneously, the combination of PAC and OC has been proposed. In the present study, we also added additives to improve the stability and biocompatibility of cyanoacrylate as a bio-glue. The purpose of this study was to evaluate the antimicrobial activity against some of the most common microorganisms associated with wound inflammation [19,20] and cellular cytotoxicity of single PAC and the combination of PAC and OC.

In addition, the use of commonly available cyanoacrylate adhesives has an unresolved major problem: the heat released during the exothermic polymerization reaction might cause the degradation of their alkyl chains into cyanoacetate and a poisonous compound formaldehyde [21]. Therefore, formaldehyde, one of the main toxic chemicals released by the degradation of PAC and OC, was quantitatively examined in this study. 

## 2. Materials and Methods

### 2.1. Chemicals

PAC (PERMABOND, Pottstown, PA, USA) and OC (EVOBOND, TONG SHEN EN-TERPRISE, Kaohsiung City, Taiwan) were purchased for the preparation of new adhesive materials. PAC was prepared by heating PAC at 145 °C for 30 min [15]. Six kinds of skin adhesive samples, such as OC, PAC, PAC7, PAC4, and PAC3, were prepared by formulating different amounts of PAC, OC, and additives. The additives include myristic acid (0–1%) and dibutyl sebacate (10–30%) as plasticizers and butylated hydroxyanisole (less than 0.01%) as an antioxidant. These ingredients in the additives are also used in the existing bioadhesives, for example, Dermabond® (Ethicon Inc., a subsidiary of Johnson & Johnson, Bridgewater, NJ, USA), and the relative proportions of each component are an important factor that determines the bonding characteristics of each company’s product. The compositions and formulating ratios of OC, PAC, and the additives are described in Table 1.

### 2.2. Cells

Testing bacteria, including *Escherichia coli, Pseudomonas aeruginosa*, and *Staphylococcus aureus*, and a testing fungus, *Candida albicans,* were purchased from Korean Cell Line Bank (Jongno-gu, Seoul, Republic of Korea). BBL^TM^ Mueller Hinton Broth (Spectrum Chemical MFG Co., New Brunswick, NJ, USA) and DIFCO^TM^ Nutrient agar (BD, Becton Drive Franklin Lakes, NJ, USA) were used to cultivate bacteria and fungus and to perform antimicrobial susceptibility testing. To prepare the culture media, 11 g of Mueller–Hinton Broth and 11.5 g of Nutrient agar were individually dissolved in 500 mL of distilled water. After that, they were sterilized by autoclaving at 120 °C for 15 min. In the case of Nutrient agar, the sterilized medium solution was solidified in Petri dishes before use. A mouse L929 fibroblastic cell line was also purchased from Korean Cell Line Bank. L929 cells were grown in Roswell Park Memorial Institute medium (RPMI; Welgene, Daegu, Republic of Korea) containing 10% fetal bovine serum (FBS, Gibco^®^, ThermoFisher Scientific, Waltham, MA, USA) and 1% penicillin–streptomycin solution (Welgene) at 37 °C in a 5% CO_2_ incubator.

### 2.3. Disc Diffusion Test for Antimicrobial Activity

Stainless plates were rigorously cleaned before use and washed with copious amounts of double-distilled water. Films (pellets, dimension Φ = 8 mm) were prepared by drying each adhesive sample of 20 μL on a paper disk (Φ 8~10 mm), allowing it to solidify in sterilized condition for 3 days and then separating it from the disk. The antimicrobial activities of six skin adhesive samples were evaluated for *Escherichia coli*, *Pseudomonas aeruginosa*, *Staphylococcus aureus*, and *Candida albicans* using the disk diffusion test. A pellet in the form of film and a 20 μL liquid droplet of each adhesive sample was carefully applied on one agar plate containing each microorganism, as shown in Figure 1. The agar plate was then incubated at 35 °C for 18–20 h. All plates were divided into three equal areas. The pellet and drop were located separately in two different areas and the remaining area was left untreated as a control. We then measured the diameter of the hollow zone around each droplet or pellet. The diameter of each droplet was also measured after each testing adhesive sample had hardened.

### 2.4. MTT (3-(4,5-dimethylthiazol-2-yl)-2,5-diphenyltetrazolium Bromide) Assay for Cytotoxicity Assessment

For the in vitro cytotoxicity assay, cells in a logarithmic growth phase were seeded on a 6-well plate at a density of 4.5 × 10^5^ cells per well. The liquid samples of PAC7 and PAC4 were spread onto slide glass and allowed to solidify for 24 h to form thin films. The films of each sample on the slides were then immersed in RPMI for 24 h at 37 °C in a 5% CO_2_ incubator to prepare the eluate. The eluate was then centrifuged for 2 min at 3000 rpm to separate and remove any detached film and its tiny fragments. The MTT assay was performed by adding MTT (Sigma-Aldrich Chemicals, St. Louis, MO, USA) to each well for a final concentration of 0.5 mg/mL. The plates were then incubated for 3 h at 37 °C. After incubation, both the MTT solution and media were carefully removed from each well and 1 mL of dimethyl sulfoxide (DMSO, Daejung^®^, Duksan General Science, Seoul, Republic of Korea) was added. The optical density (OD) value at a wavelength of 570 nm of each well was then recorded using a plate reader. Wells, including samples without cells, were used as blank. Culture medium alone was used as a negative control. Doxorubicin was used as a positive control. The graph was plotted by taking samples in the X-axis and relative cell viability value in the Y-axis.

### 2.5. Determination of Formaldehyde Content in Skin Adhesive Samples

The concentration of formaldehyde in the adhesive samples was measured according to Lynch’s protocol [22]. Briefly, the adhesive samples were dried on a glass slide, which was then placed in PBS (Welgene) solution for 24 h at 37 °C in a shaking incubator. A dilution solution was added to each PBS solution containing formaldehyde eluted from the adhesive sample for 24 h, and then the OD was measured at 410 nm using ultraviolet spectroscopy.

### 2.6. Statistical Analysis

Data are expressed as the means ± standard deviation (SD). All measurements were performed by an independent investigator who was blinded to the experimental conditions. Differences within or between normally distributed data were analyzed using a one-way ANOVA followed by Tukey’s post hoc test using Origin software (Origin Lab, Northampton, MA, USA). Differences with *p* values of less than 0.05 were considered statistically significant.

## 3. Results

### 3.1. Antimicrobial Activity of PAC in Different Concentration

The antibiotic activity of the microbial of skin adhesive samples was assessed for *Escherichia coli, Pseudomonas aeruginosa, Staphylococcus aureus*, and *Candida albicans* (Table 2 and Table 3).

Figure 1 shows the schematic illustration of the measurement of the inhibition ring in the disk diffusion test. For the droplet of the adhesive sample on the surface of the agar plate, a ring-shaped area of dehydration occurred surrounding the drop tested, and this area did not support bacterial growth. We called this area a dehydrated zone. These phenomena were observed in all the agar plates tested and thus were considered non-specific. The “dehydrated zone” refers to the phenomenon in which the surrounding area becomes slightly dry as a liquid skin adhesive sample is dropped directly onto a substrate and hardens. On the other hand, this dehydrated zone does not occur in the case of a pre-dried film (pellet). The microbial inhibition in the dehydrated zone was considered a result of the physical properties of adhesive samples irrespective of their antimicrobial activities. A ring-shaped inhibition zone that appeared the day after the experiment meant that the bactericidal activity of the adhesive sample continued. To confirm that there was no microbial growth in the inhibitory zone, a cotton swab rubbed on the surface of the area was dispersed in the medium and cultured for 20 h, but no microbial growth was observed. Unlike the results of the drop of the adhesive sample, the adhesive sample in the form of pellet showed directed inhibition rings without forming a dehydrated zone (‘Pellet’ in Figure 1.). In summary, our experiment showed that PAC, PAC7, PAC4, and PAC3 inhibited the growth of the most common microorganisms of Gram-positive, Gram-negative, and eukaryotic yeast cells, which is consistent with a previous observation that used the Kirby–Bauer disk diffusion method [23].

Table 2 and Table 3 summarize the inhibitory effects of adhesive samples in the form of a drop or pellet on the growth of each microorganism. Although the antimicrobial activity of PAC and OC mixture was clearly observed, there is no evidence that OC alone has an antimicrobial effect because the ring of the inhibition zone was not observed in the agar plate treated with the OC adhesive sample. The agar plates treated with PAC and PAC7 only showed an inhibition zone in all the plates of *Escherichia coli*. The adhesive samples containing 30% PAC (PAC3) and 100% OC (OC) were significantly less effective than those containing other concentrations of PAC or OC in the agar plates of *Escherichia coli*, *Staphylococcus aureus*, and *Candida albicans.* As shown in Table 2, the adhesive sample of PAC4 showed higher antimicrobial activity than those of OC and PAC3, and thus PAC7 and PAC4 were selected for further experiments.

### 3.2. In Vitro Cytotoxicity of Skin Adhesive Samples

To verify the cytotoxicity of skin adhesive samples, each adhesive sample mixture was added to L929 cells, a mouse fibroblast cell line, and the cell morphology and viability were measured using a microscope examination and MTT assay, respectively. An adhesive sample of PAC7 or PAC4 showed no cytotoxicity against L929 cells. Almost all the L929 cells treated with the extract of PAC7 or PAC4 maintained normal morphology compared with vehicle (culture medium)-treated cells as a negative control, as shown in Figure 2A. The results of the cytotoxicity assay were also coincident with those of the cell morphology (Figure 2B). The MTT assay indicated similar relative percentages of cell viabilities between a vehicle (medium) control and PAC7 or PAC4 adhesive sample, and there was a significant difference between PAC7 and PAC4 as a *p* value < 0.05.

### 3.3. Quantitative Analysis of Formaldehyde from Skin Adhesive Samples

The amount of formaldehyde in PBS solution in which each skin adhesive sample was placed for 24 h was measured using an ultraviolet spectroscopic method. The formaldehyde concentrations of PAC7 and PAC4 were 0.026 ppm and 0.017 ppm, respectively (Table 4). According to Korea’s National Institute of Environmental Research in 2015, the permissible exposure limit of the formaldehyde concentration in the medical device industry is below 20 ppm. Because the concentrations in both PAC7 and PAC4 were far below this concentration limit, PAC7 and PAC4 must be safe materials for manufacturing a skin adhesive sample in terms of formaldehyde release.

## 4. Discussion

Previous research has demonstrated that open wounds can often become contaminated by various microorganisms from the surrounding environment, even following disinfection and dressing treatments [13]. To solve this problem, a new biomimetic skin adhesive with improved biocompatibility, which also maintained a high level of antimicrobial activity against bacteria and fungi, was developed by using PAC, OC, and several additives.

PAC was chosen due to its rapid curing speed, while OC was selected for its excellent adhesion and durability properties, as well as its reported antimicrobial activity [24,25]. While there is currently no literature on the antimicrobial activity of PAC in the context of biomimetic skin adhesives, our results suggest that it has significant potential as a material for wound care applications due to its favorable antimicrobial properties, safety, and product quality. By combining it with OC in the right proportions, we were able to achieve optimal antimicrobial activity and biocompatibility, resulting in a biomimetic skin adhesive with a superior overall performance. These findings may have important implications for the development of improved wound care products with enhanced antimicrobial activity and biocompatibility.

The development of a new biomimetic skin adhesive that combines the advantages of both materials could potentially result in a product with high biocompatibility and antimicrobial activity while minimizing side effects. By optimizing the properties of each material, it may be possible to create a product that provides fast curing, strong adhesion, and effective antimicrobial properties without causing significant tissue irritation or other adverse effects. There is still a focus on finding an adhesive that can effectively prevent infection while minimizing scarring, reducing side effects, and improving its compatibility with biological systems.

In 1966, Ralph et al. published a paper in which the interactions between bacteriotoxin and the chain length of alkyl 2-cyanoacrylates were described. Cyanoacrylate polymers tend to increase the antibacterial effect in the order in which the chain length decreases [26,27]. In line with these works, the polymerization of ethyl cyanoacrylate seems to enhance the antibacterial effect on *Streptococcus pneumonia*, *Escherichia coli*, and *Staphylococcus aureus*. In the case of N-butyl-cyanoacrylate, the polymerization reaction increased its antibacterial effects only for Gram-positive bacteria such as *Staphylococcus aureus* and *Streptococcus pneumoniae*, but not for Gram-negative bacteria such as *Escherichia coli* and *Pseudomonas aeruginosa* [14,28]. Our results also coincide with these previous studies. Our formulations, such as PAC7 and PAC4, exhibit even better antimicrobial activity for some Gram-negative bacteria such as E. coli and *Candida albicans*, a eukaryotic yeast [29]. In the disc diffusion test, the inhibition zones formed by exposure to PAC were increased in every agar plate of *Escherichia coli*, *Staphylococcus aureus*, and *Candida albicans*. Our results, however, imply that the shorter the alkyl-chain of cyanoacrylate, the more unstable it is and the faster its chemical degradation compared to its longer alkyl-chain counterparts [30]. Several previous findings also indicated that *Escherichia coli*, a Gram-negative bacterium, was more sensitive to alkyl 2-cyanoacrylates than *Staphylococcus aureus,* a Gram-positive bacterium, because Gram-positive bacterial cells usually have thicker and more compact cell walls, and thus a more stable structure than Gram-negative bacteria [31].

So far, however, there has been little discussion about the antimicrobial effect of cyanoacrylate on Gram-positive bacteria. Rushbrook et al. argued that electrostatic interactions between cyanoacrylate monomers and the positively charged carbohydrate capsules of Gram-positive bacteria produce a strong negative net charge beyond electric neutralization, which might explain why OC shows bactericidal activity against Gram-positive bacteria but not Gram-negative bacteria [13]. In Table 2 and Table 3, the skin adhesive samples of PAC, PAC7, PAC4, and PAC3 showed no antimicrobial activity only for *Pseudomonas* sp., despite significantly inhibiting all other tested microorganisms such as *Escherichia coli*, *Staphylococcus aureus*, and *Candida albicans.* This may be partially attributed to differences in the testing volumes of cyanoacrylate because the volumes used in this study were not unstandardized. The reason for using an unstandardized testing volume is that the amounts of components used, such as PAC and OC, varies depending on the type of skin adhesive samples. Developing a biomimetic adhesive with excellent antimicrobial properties requires finding the optimal ratio of the two key components, PAC and OC, in skin adhesive samples to maximize their antimicrobial activity while also minimizing production costs by varying the concentrations and volumes of these components.

Furthermore, several studies have revealed that infections with *Pseudomonas aeruginosa* are difficult to treat and are often fatal [32,33,34,35]. Because of its ability to survive for a long period while adhered to surfaces, with minimum nutritional requirements and high tolerance to environmental variations, it is considered a clinically important opportunistic pathogen which is often associated with hospital infections. Biofilm formation, which is critical in its persistence and dissemination, is one of the main resistance mechanisms of *Pseudomonas aeruginosa* that allow it to survive. In addition to escaping host defense mechanisms, this bacterium also exhibits a strong resistance to antimicrobial agents. It produces two types of protein adhesions (pilus and non-pilus) that are important for the successful colonization of abiotic surfaces [36,37]. Understanding the advancements in antimicrobial treatments for pseudomonal infections is crucial as the standard first-line empirical treatment for these infections which involve the administration of at least two antipseudomonal therapies in combination. PAC is a commonly used tissue adhesive that is effective against most Gram-positive and Gram-negative bacteria. We suggest that its use for wound closure may reduce postoperative wound infection, especially for high-risk Gram-positive bacterial infections. The adhesive samples containing a 70% or 40% concentration of PAC represent a difference between the antibacterial properties associated with PAC and the initial goal of combining PAC and 2-octyl cyanoacrylate. Thus, both perspectives should be considered in future studies.

One of the most serious problems relating to skin adhesive is its decomposition, and the production and accumulation of formaldehyde and cyanoacetate can occur and cause cell death and the release of oxygen free radicals. It can contribute to the loss of tissue and promote the release of other detrimental mediators such as cyano-compounds, which exacerbate local ischemia, necrosis, and tissue damage processes. The patterns of polymer decomposition depend on the molecular weight of the shaped polymer. Methyl-2-cyanoacrylate, a reduced-molecular-weight polymer, degrades faster than the corresponding alkyl cyanoacetate, producing formaldehyde and other breakdown products [38]. On the other hand, when cyanoacrylate monomers with longer alkyl chains, such as n-butyl-2-cyanoacrylate and OC, are polymerized slowly, more flexible polymers are formed and the decomposition also proceeds, slowly producing less toxic degradation products, which is the reason that these are greater counterparts. Longer-chain cyanoacrylate monomers are therefore regarded as less toxic in comparison to their smaller side-chain counterparts due to their slower decomposition. The formation of the toxic degraded polymer, alkyl-2-cyanoacrylate, decreases as the length of the alkyl side chain (-R) and the molecular weight of the polymer increase [39,40]. Following the Korea National Institute of Environmental Research in 2015, the permissible exposure limit of formaldehyde is below 20 ppm in liquid form. The formaldehyde contents in our preparations of PAC7 and PAC4 are below the required level. Formaldehyde released into the microenvironment of the wound directly affects neighboring cells and is potentially toxic. Basically, all medical applications must be non-toxic, and it should be guaranteed that there are no harmful side effects. Therefore, we tested the in vitro cell cytotoxicity of adhesive samples containing PAC7 or PAC4. Assessments of cell viability and cytotoxicity are necessarily required in the procedure of a biocompatibility test. Skin fibroblasts are the most significant cells in the basic steps of wound healing. They are essential for angiogenesis, epithelialization, and collagen formation in the dermal layer. Many fibroblasts mature into myofibroblasts that encourage wound closure during progressive trauma healing [41,42].

The current study showed that PAC at a concentration of 70% or 40% was not toxic to human normal fibroblasts, as evaluated by an MTT assay. Previous findings of the effects of formaldehyde on cell proliferation and death showed that formaldehyde caused necrosis or apoptosis at 10.0 mm and reduced mitotic activity at 1.0 mm. It also increased cell proliferation and reduced apoptosis at 0.5 and 0.1 mm, respectively [43]. Our results for the cell cytotoxic experiment coincide with the above results for formaldehyde safety. In the present study, the amount of formaldehyde released was found to be under 0.04 mM, which indicates that our skin adhesive mixture was not toxic to skin fibroblast cells. In addition, PAC and OC in 40 % or 30 % proportions have reduced cytotoxicity compared to other combinations. Researchers have previously demonstrated that PAC creates longer-chain structures, resulting in the enhanced biocompatibility of prevalent cyanoacrylates. However, as shown in Figure 2., probably owing to the enhanced alkyl side chain of OC, the formation of toxic degraded polymers is restricted so that the greater proportion of OC shows reduced cytotoxicity.

In conclusion, the mixture of PAC, OC, and additives at a ratio (%) of 70:0:30 (PAC7) or 40:30:30 (PAC4) resulted in maximum antimicrobial activity and minimum formaldehyde degradation and cell toxicity among the various formulations we tested, which means that these formulations are highly effective for skin closure materials. We therefore expect that these new combinations of PAC/OC/additives will be invaluable for developing an alternative tissue adhesive with standard suture products and/or commercial tissue adhesives for skin wound closure. The cytotoxic effects of PAC in vitro in keratinocytes, in tissue culture, and in animal models should be demonstrated in the future.

## Figures and Tables

**Figure 1 materials-16-03427-f001:**
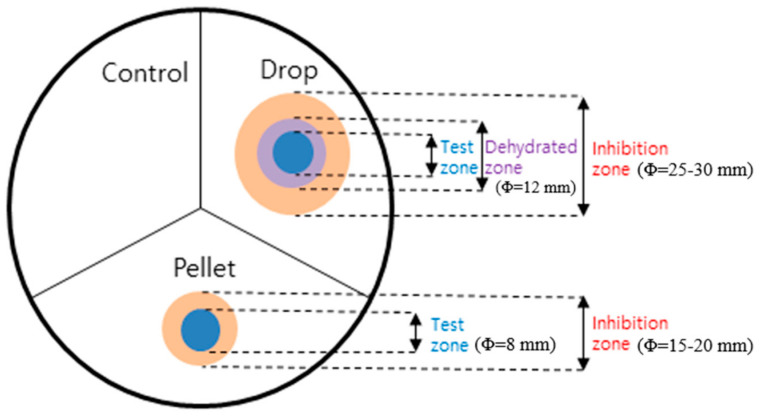
Graphical summary of measuring antimicrobial activities in the disc diffusion test. The colored areas of the ring-shaped region are distinct from one another, with the blue color indicat-ing the test zone, purple indicating the dehydrated zone, and red indicating the inhibition zone.

**Figure 2 materials-16-03427-f002:**
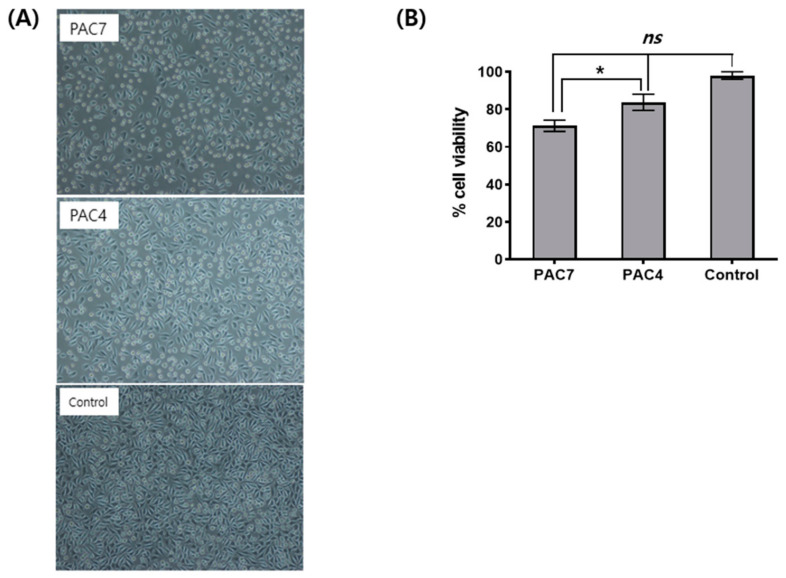
In vitro cytotoxicity tests of PAC7 and PAC4 in L929 cells. Cell morphology (**A**) was observed after indirect treatments of PAC7 and PAC4. Cell viability (**B**) was determined using MTT assay. The results are presented as means ± SDs of triplicate samples. *ns*: not significant. * *p* < 0.05 vs. the doxorubicin control.

**Table 1 materials-16-03427-t001:** Names, compositions, and formulating ratios of skin adhesive samples such as OC, PAC, PAC7, PAC4, and PAC3.

Sample Name	Composition and Formulating Ratios		
	2-Octyl Cyanoacrylate (OC)	Prepolymer Allyl 2-Cyanoacrylate (PAC)	Additives
OC	100%	0%	0%
PAC	0%	100%	0%
PAC7	0%	70%	30%
PAC4	30%	40%	30%
PAC3	40%	30%	30%

**Table 2 materials-16-03427-t002:** Effects of skin adhesive samples such as OC, PAC, PAC7, PAC4, and PAC3 on microbial growths observed around the droplet of each adhesive sample.

Microorganisms	Skin Adhesive Samples				
	OC	PAC	PAC7	PAC4	PAC3
	Droplet	Droplet	Droplet	Droplet	Droplet
*Escherichia coli*	−	+	+	−	−
*Pseudomonas aeruginosa*	−	−	−	−	−
*Staphylococcus aureus*	−	++	++	+++	+
*Candida albicans*	−	+	+	++	+

+: low inhibition (≤1 mm in radius on one side of the inhibition zone). ++: medium inhibition (3~5 mm in radius on one side of the inhibition zone). +++: high inhibition (≥5 mm in radius on one side of the inhibition zone). −: no inhibition.

**Table 3 materials-16-03427-t003:** Effects of skin adhesive samples such as OC, PAC, PAC7, PAC4, and PAC3 on microbial growths observed around the pellet (in the form of film) of each adhesive sample.

Microorganisms	Skin Adhesive Samples				
	OC	PAC	PAC7	PAC4	PAC3
	Pellet	Pellet	Pellet	Pellet	Pellet
*Escherichia coli*	−	+	+	−	−
*Pseudomonas aeruginosa*	−	−	−	−	−
*Staphylococcus aureus*	−	++	++	++	+
*Candida albicans*	−	+	+	+	+

+: low inhibition (≤1 mm in radius on one side of the inhibition zone). ++: medium inhibition (3~5 mm in radius on one side of the inhibition zone). −: no inhibition.

**Table 4 materials-16-03427-t004:** The concentrations of formaldehyde secreted from PAC7 and PAC4 adhesive samples.

	PAC7	PAC4
Formaldehyde conc. (ppm)	0.026	0.017

## Data Availability

Not applicable.

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
