# Peer review of "Antimicrobial Activity and Cytotoxicity of Prepolymer Allyl 2-cyanoacrylate and 2-Octyl Cyanoacrylate Mixture Adhesives for Topical Wound Closure"

_materials, 2023, doi:10.3390/ma16093427_

Round 1

Reviewer 1 Report

The work described in the manuscript “Antimicrobial Activity and Cytotoxicity of Prepolymer Allyl 2-2 cyanoacrylate and 2-Octyl Cyanoacrylate Mixture Adhesives for 3 Topical Wound Closure” is relevant and properly conceived.

However, the presentation of the results is not very clear therefore I suggest some major changes to be publishable:

Line 157, page 4: Explain why you call it a dehydrated zone.  Is there any reason why? If it is something that does not interfere with the results why speak about it?

Pag 5, Table 2and 3 and all the results explanation:

Please try to explain the results in a clearer way. The legend:

+ : low resistance; ++ : medium resistance; +++ : high resistance; - : no efficiency 174 is confusing.

If – is no efficient it means that the bacteria will grow near the material meaning it is very resistant.

Is confusing because + means what? That there is inhibition or not?

Because if there is inhibition it means the material works meaning that there is no resistance. However, if you say that +++ is more efficient it means that the material is more active in terms of antimicrobial properties. But that is not having high resistance. High resistance means that the bacteria will grow better.

If you have activity you have a halo meaning no resistance. So, please write this part to make sure that it is clear.

Some details can improve the manuscript so I suggest some minor changes.

Line 86; Pag 2: Please rephrase the title Chemical and Cells or divide it.

Line 122, Pag 3: In vitro must be in italics.

Line 125, Page 3: The 2 in CO2 must be underscored. Also, throughout the entire manuscript, the symbol “°” of 35°C is not properly written.

Author Response

Response to Review Reports

Reviewer 1

The work described in the manuscript “Antimicrobial Activity and Cytotoxicity of Prepolymer Allyl 2-2 cyanoacrylate and 2-Octyl Cyanoacrylate Mixture Adhesives for 3 Topical Wound Closure” is relevant and properly conceived. However, the presentation of the results is not very clear therefore I suggest some major changes to be publishable:

Line 157, page 4: Explain why you call it a dehydrated zone. Is there any reason why? If it is something that does not interfere with the results why speak about it?

Response)  In Fig. 1., the ‘dehydrated zone’ refers to the phenomenon where the surrounding area becomes slightly dry as a liquid skin adhesive sample is dropped directly onto a substrate and hardens. On the other hand, this dehydrated zone does not occur in the case of a pre-dried film (pellet). Therefore, Figure 1 shows this difference to explain the distinction between the two forms (droplet and film) of skin adhesive samples when applied to the agar plates for testing antimicrobial activity. For a better understanding, we added the paragraph in the revised manuscript as follows (line 162-165 in page 8): “ the "dehydrated zone" refers to the phenomenon where the surrounding area becomes slightly dry as a liquid skin adhesive sample is dropped directly onto a substrate and hardens. On the other hand, this dehydrated zone does not occur in the case of a pre-dried film (pellet).”

Pag 5, Table 2and 3 and all the results explanation:

Please try to explain the results in a clearer way. The legend:

+ : low resistance; ++ : medium resistance; +++ : high resistance; - : no efficiency 174 is confusing.

If – is no efficient it means that the bacteria will grow near the material meaning it is very resistant.

Response) As the reviewer mentioned, “no efficient” means that adhesive sample did not exhibit any inhibition on the target microorganism, so the growth of the microorganism on the plate was normal.

Is confusing because + means what? That there is inhibition or not?

Response)  “+” means the level of inhibition on the growth of target microorganism on the agar plate. So it was meant that the more ‘+’, the worse the growth is. For a better understanding, ‘resistance’ was changed with ‘inhibition’ in Table 2 and 3 in the revised manuscript. 

Because if there is inhibition it means the material works meaning that there is no resistance. However, if you say that +++ is more efficient it means that the material is more active in terms of antimicrobial properties. But that is not having high resistance. High resistance means that the bacteria will grow better.

If you have activity you have a halo meaning no resistance. So, please write this part to make sure that it is clear.

 Response) According to the reviewer’s comment, “resistance” was changed to “inhibition” in the foot notes of Table 2 and 3 in the revised manuscript.

Some details can improve the manuscript so I suggest some minor changes.

Line 86; Pag 2: Please rephrase the title Chemical and Cells or divide it.

Response) According to the reviewer’s comment, a section of “2.1. Chemicals and Cells” in Materials and Methods was divided into two parts such as “2.1. Chemicals” and “2.2. Cells”. 

Line 122, Pag 3: In vitro must be in italics.

 Response) According to the reviewer’s comment, all “in vitro” were italicized.

Line 125, Page 3: The 2 in CO2 must be underscored. Also, throughout the entire manuscript, the symbol “°” of 35°C is not properly written.

 Response) According to the reviewer’s comment, all temperature units were corrected to “ ℃ ”.

Reviewer 2 Report

The manuscript reports the antimicrobial activity and cytotoxicity of mixtures of commercially available prepolymerized cyanoacrylates and an undisclosed additive. Without information about the additive, it is knot possible to draw conclusions.

Author Response

Response to Review Reports

Reviewer 2

The manuscript reports the antimicrobial activity and cytotoxicity of mixtures of commercially available prepolymerized cyanoacrylates and an undisclosed additive. Without information about the additive, it is not possible to draw conclusions.

 Response) According to the reviewer’s comment, we added a paragraph describing the components and functions of “Additive” in Abstract in the revised manuscript and as follows (line 95-101 in page 5): “The additives include myristic acid (0-1%) and dibutylsebacate (10-30%) as a plasticizer, and butylated hydroxyanisole (less than 0.01%) as an antioxidant.”

 These ingredients used in the additives are commonly used in the existing bioadhesive, Dermabond (a product of Ethicon, a subsidiary of J&J), and the relative proportions of each component are important information that determines the bonding characteristics of each company's product. However, we did not provide detailed information on this due to it being considered confidential information of the respective companies.

Reviewer 3 Report

This investigation was aimed at developing a skin tissue adhesive by mixing prepolymer allyl 2-cyanoacrylate, 2-octyl cyanoacrylate, and additives in various proportions, and examining the antimicrobial activity, cell cytotoxicity and formaldehyde release. The authors suggested that the samples containing 70% or 40% prepolymer allyl 2-cyanoacrylate, have the highest antimicrobial activities and no cytotoxicity. No adhesive samples released formaldehyde higher than the permitted level. The study designs and methods are logistics. However, the interpretations of some results are not reasonable due to missing of antimicrobial inhibition zone data and cell viability. Many statement should also be corrected or clarified for readers. The suggestions for the authors as listed below:

Introduction

1.     P. 2, line 52: “Alkyl octyl cyanoacrylate adhesives with a longer side chain of 8 carbon atoms thus “. Octyl cyanoacrylate adhesives are those with a side chain of 8 carbon atoms. The following sentences (line 55-60) are not easily understood. Please rewrite these sentences.

2.     P. 2, line 68: please edit this sentence as: “… it is still not commonly used in medicine, as it is not well known in the industry and is still costly [16-18].”

3.     The purpose of this study was to evaluate the antimicrobial activity of the combination of PAC and 2- octyl cyanoacrylate, but in the MM section, octyl cyanoacrylate was used. Which one is correct?

M & M:

1.     What is the additives? Is it an organic or inorganic component?

2.     2.1 I would suggest that the dispensation of these experimental adhesives and their components first, then the cells and bacterial can be mentioned in their individual sections.

3.     2.2 How were the pallet fabricated? Are they the films mentioned in line 105? The reason to use both pallets and drops should be mentioned. The cultures for Escherichia coli, Pseudomonas aeruginosa, Staphylococcus aureus and Candida albicans should be mentioned. Do you mean “For each bacteria, 6 samples from each combinations were used for examination of antimicrobial activities? Please state clearly.

4.     Candida albicans is not a bacteria. Why did the authors choose it for test? How do you culture it?

5.     Line 124: what is the meaning of this sentence “sample covered slides dipped and dissolved in RPMI for 24 hours at 37â—¦C in a 5% CO2”?

Results:

1.     3.1. The figures of bacterial inhibition and the size of inhibition zones should be illustrated.

2.     Table 2 and 3: The tables only list the effects of these adhesives as “-””+”. What is the definition for resistant, medium and high resistant? Actually, resistance usually represents the bacterial resistance to antibiotic treatment. Do you mean low efficiency, medium efficient…. here?

3.     3.1  line 168: PAC5 was mentioned here, but no preparation of PAC5 was listed in materials and methods.

4.     Line 191: please correct the topic of this session.

5.     Only one-day culture for the examination of cytotoxicity might be too short. Can you provide data of 3-7 day culture?

6.     Fig. 2: Is there significantly differences between control and PAC4, PAC7?

7.     Line 215: the law for Korea’s National Institute of Environmental Research might be only applied for the manufacturers, not the patients receiving the adhesive therapy. Is there other regulations for these materials?

Discussion:

1.      P. 221: Do you mean exogenous microorganisms spread from wound dressing materials?

2.      Line 231: double “with with”.

3.      Line 230-232, these sentences should be edited.

4.      The first paragraph should be more focused on the components Allyl 2-cyanoacrylate, Octyl cyanoacrylate, and additives in terms of their antibacterial activity. The effects of side chain length, bacterial species, and polymerization should be discussed one by one.

5.      Line 251: why the authors considered the volumes of cyanoacrylate used in this study were not unstandardized?

6.      Line 298: there is no comparisons between the control and experimental groups. PAC7 and PAC4 showed significant reductions in cell numbers.

7.      Line 306-311: only PAC7 and PAC4 are shown in the cell viability. These discussions about the proportion of PAC and 2-octyl cyanoacrylate could be too much. 

M & M:

1.     What is the additives? Is it an organic or inorganic component?

2.     2.1 I would suggest that the dispensation of these experimental adhesives and their components first, then the cells and bacterial can be mentioned in their individual sections.

3.     2.2 How were the pallet fabricated? Are they the films mentioned in line 105? The reason to use both pallets and drops should be mentioned. The cultures for Escherichia coli, Pseudomonas aeruginosa, Staphylococcus aureus and Candida albicans should be mentioned. Do you mean “For each bacteria, 6 samples from each combinations were used for examination of antimicrobial activities? Please state clearly.

4.     Candida albicans is not a bacteria. Why did the authors choose it for test? How do you culture it?

5.     Line 124: what is the meaning of this sentence “sample covered slides dipped and dissolved in RPMI for 24 hours at 37â—¦C in a 5% CO2”?

Results:

1.     3.1. The figures of bacterial inhibition and the size of inhibition zones should be illustrated.

2.     Table 2 and 3: The tables only list the effects of these adhesives as “-””+”. What is the definition for resistant, medium and high resistant? Actually, resistance usually represents the bacterial resistance to antibiotic treatment. Do you mean low efficiency, medium efficient…. here?

3.     3.1  line 168: PAC5 was mentioned here, but no preparation of PAC5 was listed in materials and methods.

4.     Line 191: please correct the topic of this session.

5.     Only one-day culture for the examination of cytotoxicity might be too short. Can you provide data of 3-7 day culture?

6.     Fig. 2: Is there significantly differences between control and PAC4, PAC7?

7.     Line 215: the law for Korea’s National Institute of Environmental Research might be only applied for the manufacturers, not the patients receiving the adhesive therapy. Is there other regulations for these materials?

Discussion:

1.      P. 221: Do you mean exogenous microorganisms spread from wound dressing materials?

2.      Line 231: double “with with”.

3.      Line 230-232, these sentences should be edited.

4.      The first paragraph should be more focused on the components Allyl 2-cyanoacrylate, Octyl cyanoacrylate, and additives in terms of their antibacterial activity. The effects of side chain length, bacterial species, and polymerization should be discussed one by one.

5.      Line 251: why the authors considered the volumes of cyanoacrylate used in this study were not unstandardized?

6.      Line 298: there is no comparisons between the control and experimental groups. PAC7 and PAC4 showed significant reductions in cell numbers.

7.      Line 306-311: only PAC7 and PAC4 are shown in the cell viability. These discussions about the proportion of PAC and 2-octyl cyanoacrylate could be too much.

Author Response

Response to Review Reports

Reviewer 3

This investigation was aimed at developing a skin tissue adhesive by mixing prepolymer allyl 2-cyanoacrylate, 2-octyl cyanoacrylate, and additives in various proportions, and examining the antimicrobial activity, cell cytotoxicity and formaldehyde release. The authors suggested that the samples containing 70% or 40% prepolymer allyl 2-cyanoacrylate, have the highest antimicrobial activities and no cytotoxicity. No adhesive samples released formaldehyde higher than the permitted level. The study designs and methods are logistics. However, the interpretations of some results are not reasonable due to missing of antimicrobial inhibition zone data and cell viability. Many statement should also be corrected or clarified for readers. The suggestions for the authors as listed below:

Introduction

  1. 2, line 52: “Alkyl octyl cyanoacrylate adhesives with a longer side chain of 8 carbon atoms thus “. Octyl cyanoacrylate adhesives are those with a side chain of 8 carbon atoms. The following sentences (line 55-60) are not easily understood. Please rewrite these sentences.

    Response) According to the reviewer’s comment, we deleted “longer” from the sentence (line 62 in page 4).

  1. 2, line 68: please edit this sentence as: “… it is still not commonly used in medicine, as it is not well known in the industry and is still costly [16-18].”

Response) According to the reviewer’s comment, we corrected it in the revised manuscript (line 75-77 in page 4) as follows : “Despite its ability to enhance the mechanical properties of the adhesive, such as its bonding strength, its usage in medicine is still limited due to a lack of awareness in the industry and its high cost.”.

  1. The purpose of this study was to evaluate the antimicrobial activity of the combination of PAC and 2- octyl cyanoacrylate, but in the MM section, octyl cyanoacrylate was used. Which one is correct?

 Response) 2- octyl cyanoacrylate is correct, so we changed “octyl cyanoacrylate” to “2- octyl cyanoacrylate” in the revised manuscript.

M & M:

  1. What is the additives? Is it an organic or inorganic component?

Response) This comment was answered in the response to Reviewer 2’s comment.

  1. 1 I would suggest that the dispensation of these experimental adhesives and their components first, then the cells and bacterial can be mentioned in their individual sections.

Response) According to the reviewer’s comment, we described six skin adhesive samples and their formulations later in “2.1. Chemicals” section in Materials and Methods in the revised manuscript (line 95-98 in page 5), and “Chemicals” and “Cells” were also described in different sections in the revised manuscript.

  1. 2 How were the pallet fabricated? Are they the films mentioned in line 105? The reason to use both pallets and drops should be mentioned. The cultures for Escherichia coliPseudomonas aeruginosaStaphylococcus aureus and Candida albicans should be mentioned. Do you mean “For each bacteria, 6 samples from each combinations were used for examination of antimicrobial activities? Please state clearly.

    Response) We added the paragraph into M&M in the revised manuscript (line 119-121 in page 6) as follows: “Films (pellets, dimension Φ=8 mm) were prepared by drying each adhesive sample of 20 ml on a paper disk (Φ 5~8mm), allowing it to solidify in sterilized condition for 3 days and then separating it from the disk.”, (line 119-121 in page 6) and “A pellet in the form of film and a 20 ml liquid droplet of each adhesive sample was carefully applied on one agar plate containing each microorganism.” (line 123-125 in page 6).

  1. Candida albicans is not a bacteria. Why did the authors choose it for test? How do you culture it?

Response) We chose Candida albicans as a testing fungus in this study. The media and culture conditions were described in “2.2. Cell” section in the revised manuscript (line 106-112 in page 6).

  1. Line 124: what is the meaning of this sentence “sample covered slides dipped and dissolved in RPMI for 24 hours at 37â—¦C in a 5% CO2”?

   Response) For a better understanding, the sentence was rewrite as follows: “The liquid samples of PAC7 and PAC4 were spread onto slide glass and allowed to solidify for 24 hours to form thin films. The films of each sample on the slides were then immersed in RPMI for 24 hours at 37℃ in a 5% CO2 incubator to prepare the eluate. The eluate was then centrifuged for 2 minutes at 3000 rpm to separate and remove any detached film and its tiny fragments.” In the revised manuscript (line 134-138 in page 7).

Results:

  1. 1. The figures of bacterial inhibition and the size of inhibition zones should be illustrated.

Response) We added them into the Fig. 1.

  1. Table 2 and 3: The tables only list the effects of these adhesives as “-””+”. What is the definition for resistant, medium and high resistant? Actually, resistance usually represents the bacterial resistance to antibiotic treatment. Do you mean low efficiency, medium efficient… here?

Response) To minimize confusion of description about the antimicrobial activity, “resistance” was changed to “inhibition” in the foot notes of Table 2 and 3 in the revised manuscript.

  1. 3.1  line 168: PAC5 was mentioned here, but no preparation of PAC5 was listed in materials and methods.

Response) We deleted ‘PAC5’ from the sentence in the revised manuscript.

  1. Line 191: please correct the topic of this session.

Response) We corrected the title of this session with “3.2. In Vitro Cytotoxicity of Skin Adhesive Samples” in the revised manuscript.

  1. Only one-day culture for the examination of cytotoxicity might be too short. Can you provide data of 3-7 day culture?

Response) Unfortunately, we do not have the data of longer incubating time of sample treatments. However, we observed that there was no difference between a one-day culture and a longer culture (2 days) in terms of cell shape and color of cell culture medium. 

  1. Fig. 2: Is there significantly differences between control and PAC4, PAC7?

Response) As we described in the manuscript, there were low levels of growth inhibition of E. coli only in PAC and PAC7. And any adhesive samples did not exhibit growth inhibition against P. aeruginosa. PAC, PAC7, PAC4and PAC3 showed noticeable inhibitory activities against both S. aureus and Candida albicans. Among these samples, PAC4 showed the best activity.  

  1. Line 215: the law for Korea’s National Institute of Environmental Research might be only applied for the manufacturers, not the patients receiving the adhesive therapy. Is there other regulations for these materials?

Response) In general, under Korean law, all medical devices can be sold below the acceptable level of formaldehyde concentration, and skin sensitization tests of film-type medical products are tested separately. However, even at this time, formaldehyde below the reference concentration does not affect the test.

Discussion:

  1. P. 221: Do you mean exogenous microorganisms spread from wound dressing materials?

Response) For a better understanding, we corrected that sentence with “Previous research has demonstrated that open wounds can often become contaminated by various microorganisms from the surrounding environment, even following disinfection and dressing treatments.” in the revised manuscript (line 216-218 in page 11).

  1. Line 231: double “with with”.

Response) According to the reviewer’s comment, we deleted it in the revised manuscript.

  1. Line 230-232, these sentences should be edited.

Response) According to the reviewer’s comment, we edited that sentence in the revised manuscript (line 245-250 in page 12) as follows : In the case of N-butyl-cyanoacrylate, the polymerization reaction increased its antibacterial effects only on Gram-positive bacteria such as Staphylococcus aureus and Streptococcus pneumoniae, but not on Gram-negative bacteria such as Escherichia coli and Pseudomonas aeruginosa [14, 26]. Our results are also coincident with these previous studies. Our formulations such as PAC7 and PAC4 are even better antimicrobial activity for some Gram-negative bacteria such as E. coli, and Candida albicans, an eukaryotic yeast.

  1. The first paragraph should be more focused on the components Allyl 2-cyanoacrylate, Octyl cyanoacrylate, and additives in terms of their antibacterial activity. The effects of side chain length, bacterial species, and polymerization should be discussed one by one.

Response) According to the reviewer’s comment, we added and corrected the first part of Discussion section in the revised manuscript (line 216-231 in page 11) as follows: “Previous research has demonstrated that open wounds can often become contaminated by various microorganisms from the surrounding environment, even following disinfection and dressing treatments. [23]. To solve this problem, a new biomimetic skin adhesive having improved biocompatibility while maintaining a high level of antimicrobial activity against bacteria and fungi was developed by using prepolymer allyl 2-cyanoacrylate (PAC), 2-octyl cyanoacrylate (OC) and several additives.

Allyl 2-cyanoacrylate was chosen due to its fast curing speed, while 2-octyl cyanoacrylate was selected for its excellent adhesion and durability properties, as well as its reported antimicrobial activity [24, 25]. While there is currently no literature on the antimicrobial activity of allyl 2-cyanoacrylate in the context of biomimetic skin adhesives, our results suggest that it has significant potential as a material for wound care applications, due to its favorable antimicrobial properties, safety, and product quality. By combining it with 2-octyl cyanoacrylate in the right proportions, we were able to achieve optimal antimicrobial activity and biocompatibility, resulting in a biomimetic skin adhesive with superior overall performance. These findings may have important implications for the development of improved wound care products with enhanced antimicrobial activity and biocompatibility.”

And we also two references 24, and 25.

  1. Line 251: why the authors considered the volumes of cyanoacrylate used in this study were not unstandardized?

Response) We added the following paragraph in to Discussion section in the revised manuscript (line 269-274 in page 13): “The reason for using an unstandardized testing volume is that the amount of components, such as PAC and OCA, used varies depending on the type of skin adhesive samples. Developing a biomimetic adhesive with excellent antimicrobial properties requires finding the optimal ratio of the two key components, PAC and OC, in skin adhesive samples to maximize their antimicrobial activity while also minimizing production costs by varying the concentrations and volumes of these components.

  1. Line 298: there is no comparisons between the control and experimental groups. PAC7 and PAC4 showed significant reductions in cell numbers.

Response) According to the reviewer’s comment, we corrected statistical significance in Fig 2(B).

  1. Line 306-311: only PAC7 and PAC4 are shown in the cell viability. These discussions about the proportion of PAC and 2-octyl cyanoacrylate could be too much. 

Response) According to the reviewer’s comment, we reduced the discussion part about the cell viability of PAC7 and PAC4 and more focusing on the relationship the toxicity of PAC7 and PAC4 with formaldehyde safety of the skin adhesive products in the revised manuscript.

Round 2

Reviewer 1 Report

 The manuscript is acceptable for publication in the present form.

Author Response

Thank you for allowing me to publish my revised manuscript on Materials.

Reviewer 2 Report

The manuscript addreses the concerns in the previous review.

Author Response

(The authors gave the same response as above.)

Reviewer 3 Report

The authors have revised this manuscript and corrected some statements. Some unclear points have also been clarified. However, some errors are generated after the editing. The strong cytotoxicity of this material constructs a big problem, which may imped its in vivo use. Please edit the following points:

Title

The authors are advised to rename the term "prepolymer allyl 2-cyanoacrylate" to "pre-polymerized allyl 2-cyanoacrylate" based upon the information provided in the reference 15 as well as the preparation method mentioned here.

Abstract

The cytotoxicity is evident, but no cytotoxicity is stated here.

Introduction

1.     Line 69: Does “OC” represent 2-octyl cyanoacrylate or octyl cyanoacrylate? The abbreviation should be quoted in its first standing place.

2.     The following sentence “A longer side chain of octyl cyanoacrylate provides various potential benefits such as greater strength and flexibility to OC adhesives than those of medium- or short-chained cyanoacrylate adhesives, including butyl cyanoacrylate adhesives.” Does this sentence focus on the comparisons of octyl cyanoacrylate and 2-octyl cyanoacrylate, or the comparisons of octyl cyanoacrylate and medium- or short-chained cyanoacrylate adhesives?

3.     Line 96: “….chains into cyanoacetate and formaldehyde, a poisonous compound [21].” Should be “….chains into cyanoacetate and a poisonous compound formaldehyde [21]”

M & M

1.     Line 139: Please correct the term “Film” as “Pellets”. The dimension is not consistent between the paper disk and the film.

Results:

1.     Table 2 &3: The definition of low, medium, and high inhibition is still missing.

Discussion

1.     The first paragraph: I do not think this tissue adhesive is a biomimetic material, especially there is significant cytotoxicity.

2.     The abbreviation as PAC, OC should be used to replace allyl 2-Cyanoacrylate and 2-octyl cyanoacrylate since they appear at the first time.

3.     Line 348: what is OCA?

Author Response

Reviewer 3’s comment

The authors have revised this manuscript and corrected some statements. Some unclear points have also been clarified. However, some errors are generated after the editing. The strong cytotoxicity of this material constructs a big problem, which may imped its in vivo use. Please edit the following points:

Title

The authors are advised to rename the term "prepolymer allyl 2-cyanoacrylate" to "pre-polymerized allyl 2-cyanoacrylate" based upon the information provided in the reference 15 as well as the preparation method mentioned here.

Abstract

The cytotoxicity is evident, but no cytotoxicity is stated here.

Response) We apologize that the statistical notation for Figure 2(B) in the 1st revised manuscript was incorrect. As stated in the original manuscript, there is no statistical significance between control vs. PAC4 and PAC7, and there is a significant difference between PCA4 and PAC7 as p value < 0.05. We therefore corrected it in Fig 2(B) in the 2nd revised manuscript.

Introduction

  1. Line 69: Does “OC” represent 2-octyl cyanoacrylate or octyl cyanoacrylate? The abbreviation should be quoted in its first standing place.

   Response) Yes, OC represents 2-octyl cyanoacrylate, and for a better understanding, “OC” was changed with “cyanoacrylate” in the 2nd revised manuscript (line 65 in page 4).

  1. The following sentence “A longer side chain of octyl cyanoacrylate provides various potential benefits such as greater strength and flexibility to OCadhesives than those of medium- or short-chained cyanoacrylate adhesives, including butyl cyanoacrylate adhesives.” Does this sentence focus on the comparisons of octyl cyanoacrylate and 2-octyl cyanoacrylate, or the comparisons of octyl cyanoacrylate and medium- or short-chained cyanoacrylate adhesives?

Response) It indicates the comparisons of a 2-octyl cyanoacrylate-containing adhesive with medium- or short-chained cyanoacrylate adhesives (line 64-66 in page 4).

  1. Line 96: “….chains into cyanoacetate and formaldehyde, a poisonous compound [21].” Should be “….chains into cyanoacetate and a poisonous compound formaldehyde [21]”

 Response) According to reviewer’s comment, it corrected into “a poisonous compound formaldehyde” (line 86 in page 5).

M & M

  1. Line 139: Please correct the term “Film” as “Pellets”. The dimension is not consistent between the paper disk and the film.

Response) According to reviewer’s comment, “Film” was removed from the sentence of line 128 in page 5, and the dimension of paper disk was corrected to Φ 8~10mm in line 120 in the 2nd revised manuscript.

Results:

  1. Table 2 &3: The definition of low, medium, and high inhibition is still missing.

Response) We added sentences describing the degree of growth inhibition in Table 2 and 3 as follows: +, £ 1 mm in radius on one side of the inhibition zone; ++, 3~5 mm in radius on one side of the inhibition zone; +++, ³ 5 mm in radius on one side of the inhibition zone, respectively.

Discussion

  1. The first paragraph: I do not think this tissue adhesive is a biomimetic material, especially there is significant cytotoxicity.

Response) As I mentioned earlier in the response to reviewer’s comment to Abstract, the statistical notation for Figure 2(B) in the 1st revised manuscript was incorrect. So we corrected it in the 2nd revised manuscript.

  1. The abbreviation as PAC, OC should be used to replace allyl 2-Cyanoacrylate and 2-octyl cyanoacrylate since they appear at the first time.

Response) According to reviewer’s comment, all terms of allyl 2-Cyanoacrylate and 2-octyl cyanoacrylate were abbreviated and highlighted in blue in the 2nd revised manuscript.

  1. Line 348: what is OCA?

Response) It was incorrectly marked by mistake. So we modified it to OC in line 266 in page 13 in the 2nd revised manuscript.
